# Bronze Age non-elite mobility in Denmark examined through a new human-based bioavailable strontium isotope range

Karin Margarita Frei[1], Malene Refshauge Beck[2], Pernille Pantmann[3], Niels Algreen Møller[4], Morten Søvsø[5], Robert Frei[6]*

1 Department of Research, Collections and Conservation, Environmental Archaeology and Materials Science, National Museum of Denmark, Denmark, 2 Vikingemuseet Ladby, Kerteminde, Denmark, 3 Museum Nordsjælland, Hillerød, Denmark, 4 Museum Thy, Thisted, Denmark, 5 Museum Vest, Ribe, Denmark, 6 Department of Geosciences and Natural Resource Management, University of Copenhagen, Copenhagen, Denmark

* robertf@ign.ku.dk

## Abstract

Strontium isotope analysis is now a key method for investigating ancient human mobility, leading to a rapid expansion of available $^{87}Sr/^{86}Sr$ datasets. Owing to the relatively homogeneous surface geological conditions across present-day Denmark (excluding Bornholm) and the growing number of regional datasets, it is now possible to construct statistically defined ranges of bioavailable strontium directly from archaeological human data. In this study, we compile 513 published strontium isotope values from tooth enamel and pars petrosa of individuals recovered from archaeological sites across present-day Denmark and add 115 new values. Using the Median Absolute Deviation (MAD) method to identify outliers in this comprehensive and diachronic database of 628 human $^{87}Sr/^{86}Sr$ ratios, we define the first statistically constrained, human-based range of bioavailable strontium isotope values for Denmark to $^{87}Sr/^{86}Sr = 0.7089–0.7117$. We interpret this range as representing typical bioavailable strontium signatures in prehistoric Denmark. We then apply it, for the first time, as one of the reference frameworks for investigating the mobility of non-elite individuals from the Nordic Bronze Age in present-day Denmark. In total, we conducted 34 strontium isotope analyses on individuals from two sites: fourteen analyses from six inhumations at Kalvehavegård on Funen, and twenty analyses from cremated individuals at Sølager on Zealand. We compare the individuals' strontium isotope values both to established baselines relevant for past mobility studies and to the new human-based range defined in this study. The results indicate that mobility during the Nordic Bronze Age was not restricted to elite social groups but also encompassed some non-elite individuals, offering new insights into social dynamics during this formative period of European prehistory. Moreover, the new strontium dataset presented here represents the first accessible, country-wide compilation of human-derived Sr data for Denmark,

**Data availability statement:** All relevant data are within the paper and its Supporting Information files.

**Funding:** This study was made possible through the funding to KMF provided by the Carlsberg Foundation "Semper Ardens" advance research grant CF18-0005 for which we are very grateful. https://www.carlsbergfondet.dk/en.

**Competing interests:** The authors have declared that no competing interests exist.

providing a robust platform for future comparative studies and mobility research in the region.

## Introduction

Over the past decade, research on mobility in Bronze Age Europe has undergone a paradigm shift, driven by scientific analyses conducted directly on human remains. In particular, the application of ancient DNA (aDNA) and strontium isotope analyses have revealed previously unanticipated levels of mobility. This shift—from an earlier assumption that human mobility was limited and rare—was catalyzed in 2015 by two major developments: a series of aDNA studies [1,2] and the first in-detailed isotopic mobility study of a Bronze Age female, the Egtved Girl, based on strontium isotope analysis [3]. These studies, now a decade old, sparked a reevaluation of long-standing assumptions and opened new avenues of inquiry into human movement during the European Bronze Age.

Subsequent research has reinforced the understanding of higher degrees of mobility, although with regional variation, as evidenced, among others, by studies in areas such as the Lech Valley in Germany and parts of Italy [4,5]. In Denmark, a large-scale study involving 88 individuals from the 3rd and 2nd millennia BCE found that, while the majority of elite individuals were of local origin, a significant number exhibited non-local strontium isotope signatures [6]. These non-local individuals included both elite males and females [6]. Further research, including analyses of cremation burials from the Late Bronze Age, confirmed that mobility remained a feature of this later period as well (e.g., [7]).

However, as highlighted by Bergerbrant and coworkers [8] in a study from present-day southern Sweden, non-elite individuals—or "commoners"—of the Nordic Bronze Age have often remained archaeologically invisible. This invisibility is due in part to the fact that the majority of excavated Bronze Age burials in the Nordic region belong to the elite. As a result, the non-elite individuals dating to the Nordic Bronze Age—despite likely constituting most of the population—are strongly underrepresented in mobility studies and in the broader interpretations of this period's social dynamics.

The present study is the first to focus explicitly on this "invisible" group, aiming to investigate their mobility and thereby contribute to a more inclusive understanding of the socio-dynamics of the Nordic Bronze Age society within the region of present-day Denmark. To this end, we conducted 34 strontium isotope analyses on 26 individuals from two Danish Bronze Age sites: Kalvehavegård and Sølager (also known as Store Karlsminde). Both sites yielded individuals buried either without grave goods or with only modest offerings and are therefore interpreted as non-elite burials. Kalvehavegård represents a recent excavation of inhumation graves situated at the edge of a burial mound on the island of Funen, near the coast, dating to the Early Bronze Age [9]. In contrast, Sølager derives from an excavation carried out more than 80 years ago, which uncovered cremation burials within a coastal mound in northern Zealand. Together, these two sites offer a rare and valuable opportunity to examine

mobility among non-elite individuals—those who have long remained understudied in the archaeological record. By doing so, this study seeks to fill a critical gap in our understanding of Bronze Age society in Scandinavia.

Yet, the effectiveness of $^{87}$Sr/$^{86}$Sr isotope analysis for identifying an individual's childhood origin depends on the availability of reliable regional baselines of bioavailable values and comparable reference data. In recent years, such baselines have been established for many parts of Europe (e.g., [10–24] and worldwide [25–30]. However, there is currently no standardized methodology for constructing reference baselines in archaeological mobility studies. Recent discussions have also highlighted potential issues related to the use of modern biosphere data to characterize past environments [16,31–34]. For example, modern agricultural contamination of soil by liming might pose a threat to the reliability of modern proxies like soil leachates and plants that draw their water, minerals and nutrients from the respective soil pore solutions. Such contamination might, in some cases, compromise their usefulness in constructing baselines that accurately and consistently reflect the biosphere conditions necessary to study the mobility of ancient humans and animals [35].

A variety of proxy materials are typically employed for the characterization of the strontium isotope biosphere signatures. These include modern and archaeological faunal remains [36–41], and modern environmental samples such as plants, soil extracts, and water [10,11,13,14,16,17,19,21–23,42–45] and many others.

The suitability of these proxies continues to be evaluated, as none appear to perfectly reflect the complete dietary $^{87}$Sr/$^{86}$Sr intake of past human populations. Their representativeness varies by region and underlying geological conditions [16,31,32]. To address the limitations of modern proxy samples in representing past biosphere conditions and dietary strontium uptake, Price and coworkers [38,46] proposed that, given a sufficiently large sample size, the human remains themselves could serve as an alternative for constructing baselines. Evans and coworkers [47] were the first to publish a summary of human strontium isotope data from a greater region available at that time. In their study they present an overview of such data in combination with oxygen isotope results from Britain, with the intention to provide a useful reference set for archaeological, forensic and environmental studies.

Our study has two complementary objectives: (1) to investigate, for the first time, the mobility of non-elite individuals dating to the Nordic Bronze Age within present-day Denmark, and (2) to establish the first country-wide, human-based reference database for bioavailable strontium within this region. To achieve this, we compile and present a comprehensive dataset of 628 strontium isotope analyses derived from archaeological human remains recovered from 159 sites across present-day Denmark (excluding the island of Bornholm; locations shown in Fig 1) and dating from the Mesolithic to the Medieval period—115 of which are new and unpublished, and 513 gathered and assembled from previously published sources. The 115 new strontium isotope compositions of archaeological human remains are from four different sites, the two Bronze Age sites that are the focus of this study (Kalvehavegård and Sølager) and additionally one Iron Age site (Søhale) and one Viking Age site (Lindegården/Ribe), all sites are highlighted in Fig 1. Data from the Iron Age and Viking Age sites are incorporated exclusively to enhance the geographical coverage of the reference dataset, and these analyses are published here for the first time.

## Materials and methods

### Samples

We conducted 34 strontium isotope analyses on human remains dating to the Nordic Bronze Age, representing a total of 26 individuals: six from Kalvehavegård and twenty from Sølager, as presented in Table 1. The Kalvehavegård samples, all derived from inhumation burials, and consist exclusively of tooth enamel. For five of the six individuals from this site, we performed multi-tooth enamel analyses—sampling from different molars (i.e., M1, M2, and/or M3)—to assess potential mobility during early life. The Sølager samples, in contrast, originate from cremation burials. As a result, the majority of these are pars petrosa samples, except for three well-preserved tooth enamel samples that remained identifiable despite the effects of cremation.

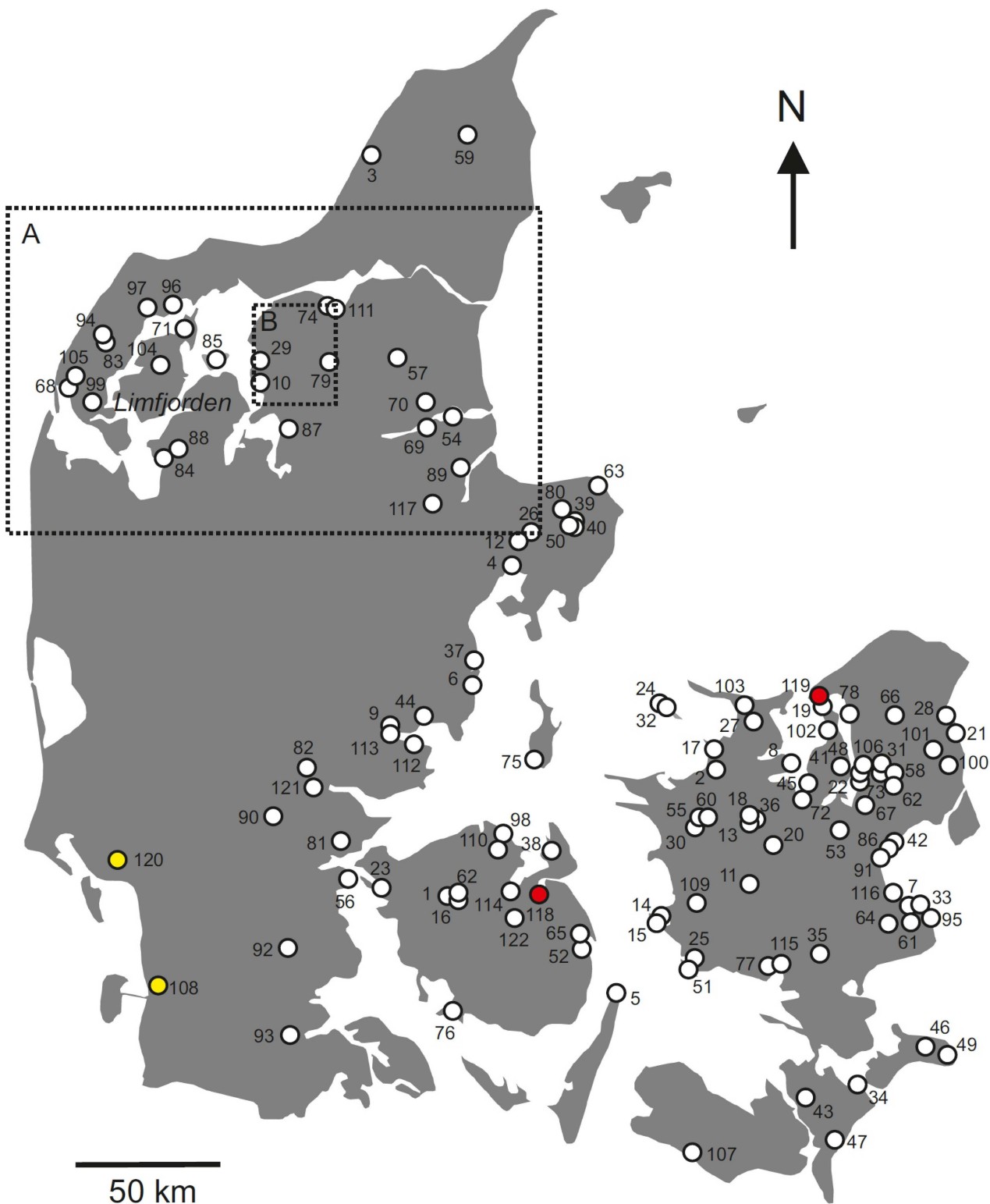

**Fig 1. Background map of Denmark with site locations.** Modified background map of Denmark (excluding Bornholm) with locations of archaeological sites from which published and new human strontium isotope values are used herein for the construction of the human-based bioavailable strontium

isotope range. Location numbers correspond to those indicated in S1 Table of the supplementary information. The sites of Kalvehavegård (nr. 119) and Sølager (nr. 118) are filled in red. Yellow filled symbols indicate the locations (Ribe, nr. 108; Søhale, nr. 120) of which the strontium isotope data are here published for the first time. The dashed outlined rectangles denote the areas in which locations from the studies of [36] (A) and [48] (B) are contained. Background map reprinted from iStock (illustration ID:927406110) under a CC BY license, with permission from rbiedermann, original copyright 2018.

For the construction of the new database of human-based bioavailable strontium isotope signatures we here only consider published $^{87}Sr/^{86}Sr$ data from human tooth enamel and/or from the otic capsule of pars petrosa bones. Respective data from Denmark were extracted from the following publications: [6,7,36,48–69]. They are summarized in S1 Table.

We complemented the published data with additional 115 new analyses, of which 81 (from Ribe and Søhale) are summarized in two internal reports by the leading author of this study to respective local museums (Museum Thy; Museum Vest) and hence published here for the first time, while 34 are new generated data exclusively for the purpose of the present study. These results are also contained in S1 Table.

There is a solid body of literature supporting the use of tooth enamel as a robust archive for strontium isotope ratios in provenance and mobility studies. Tooth enamel is valued in such research because it forms during childhood (and does not remodel like bone), is highly resistant to post-mortem alteration (diagenesis), and incorporates strontium from diet and drinking water, which reflects local environment [38,70–72].

While strontium isotope analyses in tooth enamel have been conducted for several decades, it is only within the last decade that it has become possible to conduct such analyses in cremated human remains. For example, several studies have indicated that strontium isotope analysis of the petrous portion of the temporal bone is a viable method for provenance studies of both inhumed and cremated human remains [61,73–76]. The pivotal and original study by Harvig and coworkers [61] concluded that strontium isotope ratios in the otic capsule closely align with those in dental enamel from the same individuals, regardless of whether the remains were cremated or not. This finding underscores the petrous bone's reliability in tracing childhood origins. Additionally, a more recent study by [77], which assessed the preservation of biogenic strontium isotope ratios in the otic capsule of unburnt petrous bones, supports Harvig et al.'s [61] main conclusions. These studies collectively highlight the significance of the petrous bone in archaeological provenance investigations, offering an important additional source of information.

For the purpose of the construction of the new human-based bioavailable strontium isotope database, the published and the new data are crudely categorized, as best as possible, into archaeological periods, i.e., Mesolithic (Meso), Neolithic (Neo), Bronze Age (BA), Iron Age (IA) with sub-categories Pre-Roman Iron Age (PR IA), Early Roman Iron Age (ER IA), Late Roman Iron Age (LR IA), Viking Age (VA), and Medieval (MA). As it is not the aim of the present study to discuss the archaeological context of the samples from within this database, we refer the interested reader to the respective literature indicated in S1 Table. Fig 1 provides a visual geographic overview of the sample sites on a background map of present-day Denmark, using, supplementing and modifying published site maps of [50] and [6]. Areas A and B outlined by respective rectangles in Fig 1 comprise sites of [36] and [48], and the reader is referred to these publications for the detailed site location maps.

## Archaeological sites in focus of this study

**The Bronze Age burial site of Kalvehavegård.** In 2016, a Bronze Age burial at Kalvehavegård site on the island of Funen (indicated as location nr. 118; Fig 1) was discovered. Kalvehavegård is located on the northeastern part of Funen, on the Kerteminde fjord. Through the first excavation at the site the same year, archaeologists uncovered a central elite grave containing a female and male burial in a log-coffin, similar to the famous finds from Egtved and Borum Eshøj. Although the coffin itself was only preserved as traces of decomposed wood, its form and orientation (3 meters long, 0.6 meters wide, east–west) could still be identified. It had been placed on a clean sand foundation and reinforced with large stones, with an additional stone layer covering the coffin, likely protecting it from later agricultural disturbance.

**Table 1. New strontium isotope signatures of ancient human enamel and pars petrosa from Kalvehavegård and Sølager.**

| Lab nr. | Museum/ Sample | Site | Number | Period | Grave | Sex | Age | Tooth/ Pars Petrosa | $^{87}Sr/^{86}Sr$ | 2SE | [Sr] |
|---|---|---|---|---|---|---|---|---|---|---|---|
| | Number | | on map | | | | | specifications | | | (mg/kg) |
| KF 1875−1 | ØFM 855 X 12 | Kalvehavegård | 118 | BA | Grave 12 | Male | | R. Mandible; M1 | 0.71017 | 0.00001 | |
| KF 1875−2 | ØFM 855 X 12 | Kalvehavegård | 118 | BA | Grave 12 | Male | | R. Mandible; M2 | 0.70994 | 0.00001 | |
| KF 1875−3 | ØFM 855 X 12 | Kalvehavegård | 118 | BA | Grave 12 | Male | | R. Mandible; M3 | 0.70988 | 0.00001 | |
| KF 1876−1 | ØFM 855 X13 | Kalvehavegård | 118 | BA | Grave 13 | Female | | L. Mandible; M1 | 0.70971 | 0.00001 | |
| KF 1876−2 | ØFM 855 X13 | Kalvehavegård | 118 | BA | Grave 13 | Female | | L. Mandible; M2 | 0.70960 | 0.00001 | |
| KF 1876−3 | ØFM 855 X13 | Kalvehavegård | 118 | BA | Grave 13 | Female | | L. Mandible; M3 | 0.70945 | 0.00001 | |
| KF 1878−1 | ØFM 855 A32 X 53 | Kalvehavegård | 118 | BA | Grave 32 | Female? | | R. Mandible; M1 | 0.70938 | 0.00001 | |
| KF 1878−2 | ØFM 855 A32 X 53 | Kalvehavegård | 118 | BA | Grave 32 | Female? | | R. Mandible; M2 | 0.70929 | 0.00001 | |
| KF 1879−1 | ØFM 855 A75 X54 | Kalvehavegård | 118 | BA | Grave 75 | Adolescent | | L. Mandible; M1 | 0.70945 | 0.00001 | |
| KF 1879−2 | ØFM 855 A75 X54 | Kalvehavegård | 118 | BA | Grave 75 | Adolescent | | L. Mandible; M2 | 0.70944 | 0.00001 | |
| KF 1879−3 | ØFM 855 A75 X54 | Kalvehavegård | 118 | BA | Grave 75 | Adolescent | | L. Mandible; M3 | 0.70938 | 0.00001 | |
| KF 1880−1 | ØFM 855 A77 X55 | Kalvehavegård | 118 | BA | Grave 77 | Male | | R. Mandible; M2 | 0.70949 | 0.00001 | |
| KF 1880−2 | ØFM 855 A77 X55 | Kalvehavegård | 118 | BA | Grave 77 | Male | | R. Mandible; M3 | 0.70983 | 0.00001 | |
| KF 1881−1 | ØFM 855 A78 X56 | Kalvehavegård | 118 | * | Grave 78 | Child | | Maxillar Upper R; M1 | 0.70950 | 0.00001 | |
| KF 2347 | NM 278/39 B 13306−52 | Sølager | 119 | BA | Grave 4 | Unknown | 30-45 | pars petrosa | 0.71221 | 0.00001 | 300 |
| KF 2348 | NM 278/39 B 13306−52 | Sølager | 119 | BA | Grave 7 | Female | 15-30 | pars petrosa | 0.70974 | 0.00001 | 212 |
| KF 2349 | NM 278/39 B 13306−52 | Sølager | 119 | BA | Grave 8 | n.a. | 2 | M2 lower left | 0.71027 | 0.00002 | 217 |
| KF 2350 | NM 278/39 B 13306−52 | Sølager | 119 | BA | Grave 9 | Female | 25-45 | pars petrosa | 0.71057 | 0.00001 | 120 |
| KF 2351 | NM 278/39 B 13306−52 | Sølager | 119 | BA | Grave 10 | Female | 25-45 | pars petrosa | 0.71081 | 0.00001 | 111 |
| KF 2352 | NM 278/39 B 13306−52 | Sølager | 119 | BA | Grave 11 | Unknown | 25-45 | pars petrosa | 0.71061 | 0.00001 | 128 |
| KF 2353 | NM 278/39 B 13306−52 | Sølager | 119 | BA | Grave 12 | Female? | 18-25 | pars petrosa | 0.71071 | 0.00001 | 110 |
| KF 2354 | NM 278/39 B 13306−52 | Sølager | 119 | BA | Grave 13 | Female? | 35-50 | pars petrosa | 0.71250 | 0.00001 | 119 |
| KF 2355 | NM 278/39 B 13306−52 | Sølager | 119 | BA | Grave 14 | n.a. | 4 - 6 | +3 canine upper | 0.71508 | 0.00001 | 315 |
| KF 2356 | NM 278/39 B 13306−52 | Sølager | 119 | BA | Grave 17 | Male | 35+ | pars petrosa | 0.71577 | 0.00001 | 351 |
| KF 2357 | NM 278/39 B 13306−52 | Sølager | 119 | BA | Grave 18 | n.a. | Ca. 15 | pars petrosa | 0.71136 | 0.00001 | 115 |
| KF 2358 | NM 278/39 B 13306−52 | Sølager | 119 | BA | Grave 23 | Male? | 40+ | pars petrosa | 0.71080 | 0.00002 | 124 |
| KF 2359 | NM 278/39 B 13306−52 | Sølager | 119 | BA | Grave 24 | Female? | 30-45 | pars petrosa | 0.71076 | 0.00001 | 167 |
| KF 2360 | NM 278/39 B 13306−52 | Sølager | 119 | BA | Grave 25 | Male? | 35-45 | pars petrosa | 0.71083 | 0.00001 | 104 |
| KF 2361 | NM 278/39 B 13306−52 | Sølager | 119 | BA | Grave 26 | Unknown | 35-45 | pars petrosa | 0.71048 | 0.00001 | 155 |
| KF 2362 | NM 278/39 B 13306−52 | Sølager | 119 | BA | Grave 27 | Male | 21-40 | pars petrosa | 0.71152 | 0.00001 | 208 |

*(Continued)*

**Table 1.** (Continued)

| Lab nr. | Museum/ Sample | Site | Number | Period | Grave | Sex | Age | Tooth/ Pars Petrosa | 87Sr/86Sr | 2SE | [Sr] |
|---|---|---|---|---|---|---|---|---|---|---|---|
| KF 2363 | NM 278/39 B 13306−52 | Sølager | 119 | BA | Grave 28 | n.a. | 1-2 | M1 upper left | 0.71232 | 0.00001 | 251 |
| KF 2364 | NM 278/39 B 13306−52 | Sølager | 119 | BA | Grave 29 | Female? | 30-50 | pars petrosa | 0.71060 | 0.00001 | 148 |
| KF 2365 | NM 278/39 B 13306−52 | Sølager | 119 | BA | Grave 30 | Female | 30-50 | pars petrosa | 0.71076 | 0.00001 | 106 |
| KF 2366 | NM 278/39 B 13306−52 | Sølager | 119 | BA | Grave 31a | Female? | 15-40 | pars petrosa | 0.71361 | 0.00001 | 368 |

BA = Bronze Age; * ¹⁴C dated to Late Neolithic (see [9]); n.a. = not applicable.

In the western part of the coffin, dark discoloration marked the original location of a body. However, unlike expected, the individuals had been cremated. The cremated bones were subsequently collected, wrapped, and placed in the coffin together with unburnt grave goods, which remained *in situ*. The burial dates to Period II of the Early Nordic Bronze Age (1500–1300 BCE), placing it contemporaneously with the major elite barrow burials in Jutland. It represents an early example of cremation burial practices, which began to emerge during this period, prior to becoming dominant in the Late Nordic Bronze Age (after c. 1100 BCE). The burial thus reflects a combination of new funerary customs with older traditions. Female jewelry was placed at the eastern end of the cremated bones, and on top of the bone layer additional grave goods included a finely decorated bronze sword, a twisted gold wire fibula, a small rectangular gold ingot, and two bronze nails, each 6 cm long and spaced 29 cm apart, which have been interpreted as fittings from a folding chair—an extremely rare find in Bronze Age Europe but known from Egyptian contexts. As the excavation of the richly furnished central grave at Kalvehavegård neared completion, efforts were made to estimate the original size of the burial mound. These revealed that the original diameter of the barrow was around 20 meters and that it had been constructed in the traditional manner using turf. But what makes this site very exceptional in the Danish context as well as for the purpose of the present study, is the discovery of additional flat graves of non-elite individuals just outside the northern edge of the barrow. In most cases, any additional graves would be expected to lie on the southern or eastern side of the mound, not directly to the north, making this discovery particularly unexpected. There were virtually no visible traces of grave cuts around the skeletons; only a few stones placed around them indicated the possible outlines of the graves. Without the preserved skeletal material, it is doubtful whether the graves would have been identified at all. No grave goods were associated with the burials that could assist in dating them. However, during the winter of 2017, radiocarbon dating was performed on the skeletons, revealing that the majority of these graves belonged to the Early Nordic Bronze Age period I (1700–1500 BCE), i.e., slightly older then the central grave, which is dated to period II (1500−1300 BCE) [9]. Seven well-preserved skeletons were found, and additionally some skeletal remains from four other individuals could also be identified. Hence at least 11 individuals were buried at the site prior to the construction of the barrow. The flat graves are arranged in two north–south rows and appear to be oriented in relation to the barrow. A complete cross-section of society is represented here: adults of different sexes, as well as children. The youngest individual is an infant. All individuals were laid on their backs with their heads facing west. Mollusk shells, small pieces of freshly knapped flint and potsherds were present in five of the seven well preserved graves but only two graves could be confidently identified as containing grave goods, and in both cases, these were grave goods of very modest nature. One young adult male had a bead made from a dog's canine tooth placed beside his head, while a teenage individual was buried with a necklace consisting of nearly 100 shell beads and two bone beads, a find with no direct parallels within present-day Denmark [9]. An archaeological plan drawing of the Kalvehavegård site is presented in Fig 2.

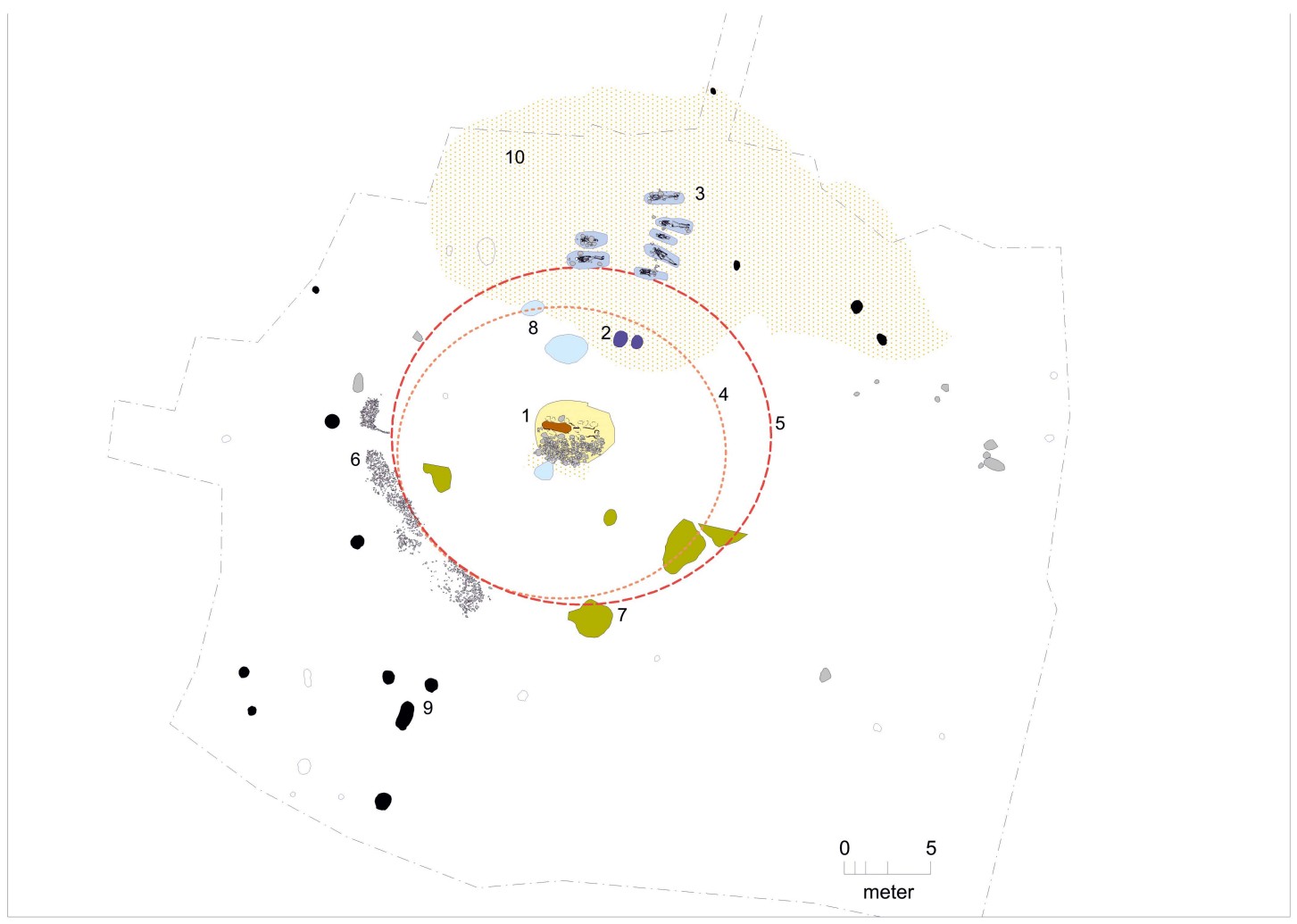

**Fig 2. Archaeological plan drawing of the Kalvehavegård site.** 1: Central grave, cremation grave in an oak coffin. 2: Two stone-set posts in a disturbed inhumation grave. 3: Burial ground with inhumation graves. 4: Preserved barrow fill. 5: Estimated original extent of the barrow. 6: Stone pavement bordering the barrow periphery. 7: Patches of burnt vegetation layer. 8: Pits containing marine mollusk shells and flint. 9: Cooking pits. 10: Bryozoan-rich sand. Graphics: Malene R. Beck.

**The Bronze Age burial site of Sølager.** Sølager (also known as Store Karlsminde) is situated on the southern coast of Halsnæs, a broad peninsula west of the town of Frederiksværk that forms the outer boundary between the inner waters of the Isefjord and the open sea of the Kattegat (location nr. 119 on Fig 1). Around 4,000 years ago, at the end of the Stone Age, the area was still an island, as sea levels were more than four meters higher than today, and a wide sound separated it from the rest of Zealand.

The cemetery at Sølager was discovered in 1939, and according to the lead excavator Becker's initial publication in 1941, 31 stone-setting burials with stone lids, dating to the Late Nordic Bronze Age, were excavated. Of these, 29 contained ceramic urns. The burials had been placed within an existing burial mound, as was customary at the time [78].

In 1979, an anthropological study (summarized in an internal report referenced to St. Karlsminde, Torup sogn, Frederiksborg amt. Mus. Nr. B 13306−52, A.S. nr. 18/77 (278/39)) was conducted on the contents of the 29 urns and concluded that each contained the cremated remains of a single individual. Of these, 24 were identified as adults and five

as children. Among the adults, 13 were assessed as female and 7 as male, while the remaining four could not be sexed (Table 1). The urns are predominantly coarse and poorly crafted, with the majority displaying a barrel-shaped profile [78].

Only a limited number of graves—approximately one-third—contained artefacts, and those that did were modest in nature. Hence, the urns served as the primary basis for dating the burials, and with the exception of one, all appear to belong to the early phase of the Late Nordic Bronze Age (1000–800 BCE). According to Becker's study [78], the graves probably represent a community of non-elite individuals from potentially a closed or short-lived settlement.

A particularly noteworthy aspect of the site of Sølager is the presence of fishhooks in two of the graves (Grave 10 and Grave 24)—a highly unusual feature, as such utilitarian tools are rarely included as grave goods [78]. The 1979 anthropological report further notes that both individuals buried in these graves were female.

Becker [78] originally interpreted the fishhooks—when considered alongside the overall modest character of the burials—as an indication that the community's subsistence strategy may have differed from that of the majority of contemporary populations. Specifically, Becker [78] proposed that the inhabitants did not rely solely on agriculture but likely supplemented their livelihood through fishing. That both graves containing fishhooks belonged to women adds a further layer of interpretive complexity and underscores the unique character of this burial site. This theory has been substantiated by findings from a recent excavation in 2006−07 of a nearby settlement Højbjerggård (NFHA 2039). In this Bronze Age settlement, bronze castings with significant metallurgical details were identified. Similar details were also identified on the bronzes from Sølager, suggesting a connection between the two sites. More importantly, a relatively significant amount of bones from cod fish found in the settlement of Højbjerggård suggests that fishing was indeed part of the socio-economy [79].

We conducted strontium isotope analyses on a total of 20 individuals from Sølager (contained in Table 1 and S1 Table), including the two female individuals with fishhooks in their graves (grave 10 and 24). An archaeological plan drawing of the Sølager site as published by [78] is presented in Fig 3.

**Supplementary archaeological sites contributing with data to the human-based strontium isotope database.** In the region of western Jutland there is a general scarcity of well-preserved human remains, compared to other regions of Denmark, which is likely due to several interrelated factors. The region's sandy, acidic soils are poorly suited for the long-term preservation of bone, leading to significant taphonomic loss over time [80,81]. As a result, only a limited number of strontium isotope datasets of humans from western Jutland have been published to date. To address this gap, in the present study we include additional 81 unpublished strontium isotope data from two sites from western Jutland (all of which have been conducted by the first author of this study). While none of these data represent individuals dating to the Bronze Age, they are included herein because of their significant contribution to the overall geographic coverage of the new and herein presented country-wide human-based bioavailable strontium isotope database for Denmark. The further archaeological interpretation of the results of these analyses is beyond the scope of the present study and therefore will be discussed elsewhere.

The Lindegården (ASR 13) site from Ribe, western Jutland (location nr. 108 on Fig 1), dates to the second half of the 9thC to the first half of the 11thC CE. The excavation trench covered an area of about 500 m² and was situated just south of Ribe Cathedral. The excavated individuals were all buried in the Christian cemetery belonging to the cathedral. The excavation was carried out in two campaigns between 2008 and 2012. Further information about the site can be found in [82]. In the present study we present the results of strontium isotope analyses of 55 individuals excavated from Lindegården site, all of which were performed on tooth enamel (S1 Table).

The Søhale site (ESM 2139; location nr. 120 on Fig 1) is an urnfield site located in western Jutland ten kilometers inland near the town of Esbjerg. The graves primarily date to the Early Preroman Iron Age. The site was excavated in 1996 and further information about the site can be found in [83] as well as in the archives of Museum Vest. The strontium isotope analyses from 26 cremated individuals are all from pars petrosa (S1 Table).

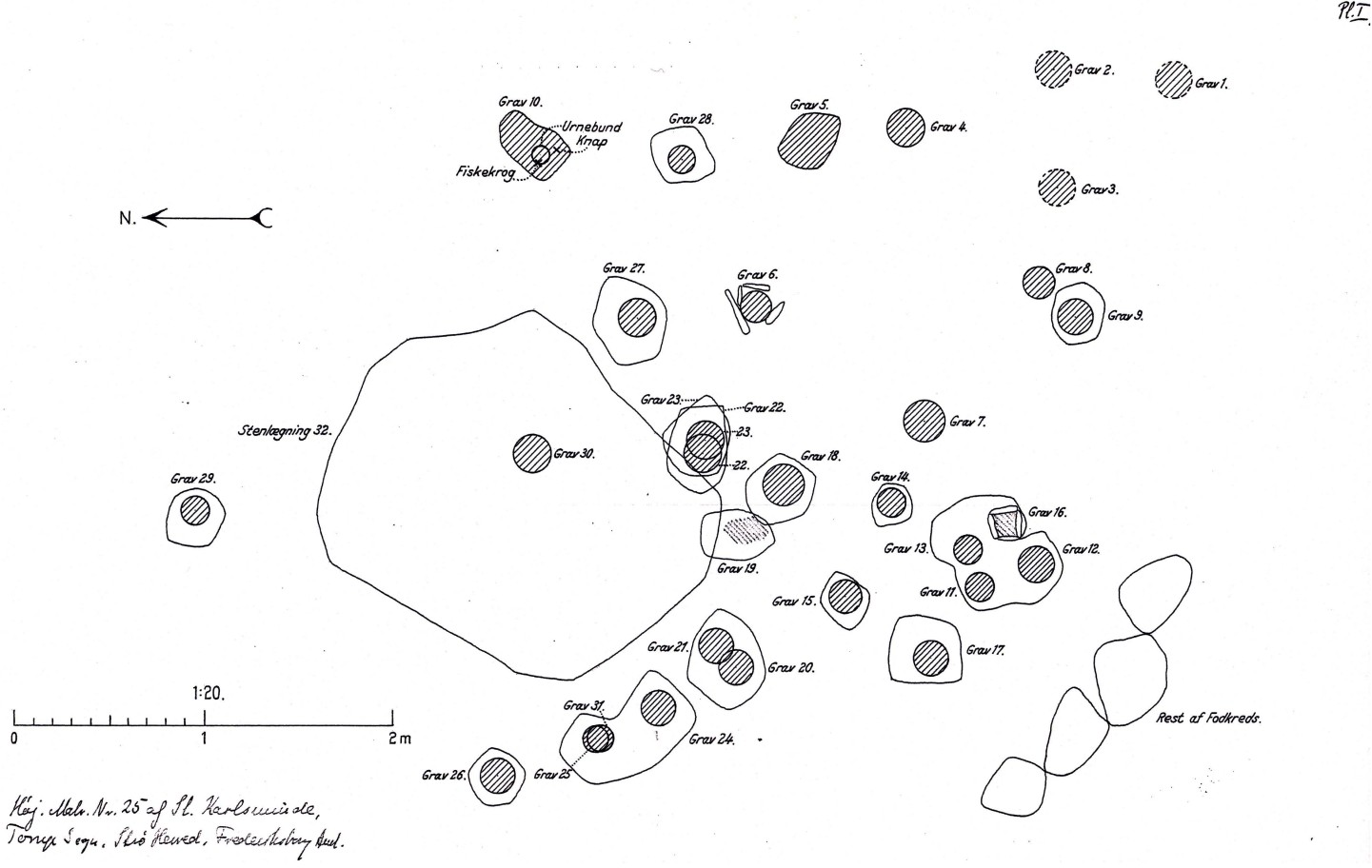

**Fig 3. Archaeological plan drawing of the Sølager site.** Original plan drawing as published by [78]. Graves 10, 24 and 28 are specifically mentioned in the text. Grave 10 and 24 are local female individuals with fishhooks added to the graves, whereas grave 28 contained a child with a non-local strontium isotope signature (for details refer to text). "Grav" is the Danish word for grave.

## Analytical techniques

Archaeological human tooth samples were first pre-cleaned by Milli-Rho-Milli-Q (MQ) $H_2O$ and subsequently by mechanical abrasion with a Dremel tool fitted with a sanding bit to remove the outer surface and potential contamination there. Small tooth enamel samples were separated with a pre-cleaned Dremel tool with a diamond bit. The samples (~2–5 mg) were dissolved in a 1:1 mixture of 30% $HNO_3$ (Seastar™) and 30% $H_2O_2$ (Seastar™). The samples typically decomposed within 15–30 min, after which the solutions were dried down on a hotplate at 80°C.

The sampling of the cremated petrous bones for Sr isotope analysis was done following the methods described by [61,84], and 1–2 mg of the densest part of the otic capsule was sampled. The samples were prepared and collected using a dental diamond drill, dissolved in pre-cleaned Teflon beakers using a 1:1 solution of 0.5 ml 6M HCl and 0.5 ml 30% $H_2O_2$, spiked with a 84Sr enriched tracer to determine the Sr concentration via isotope dilution (ID) and then dried down on a hot plate at 100°C.

1mL pipette tips were fitted with pre-cleaned, pressed-in filters to serve as disposable extraction columns. The columns were charged using 200 µl pre-cleaned SrSpec™ resin (50–100 mesh; Eichrome Inc.™/Tristchem™) conditioned with 3M $HNO_3$. The prepared samples were re-dissolved in a few drops of 3M $HNO_3$, loaded onto the columns, and washed using

~10 mL of 3M HNO$_3$. Finally, the Sr was collected using 2 mL of MQ and dried down on a hot plate at 120°. In this, the elution recipe essentially followed that by [85], scaled to our needs. Strontium was eluted/stripped by pure deionized water, and then the eluate was dried on a hot plate.

Sr column separation and thermal ionization mass spectrometry (TIMS) measurements were performed at the Department of Geoscience and Natural Resource Management (IGN) at the University of Copenhagen and largely followed the methods used by [15].

A VG Sector 54 IT mass spectrometer equipped with nine Faraday detectors was used to determine the Sr concentrations via ID protocols and Sr isotope compositions. The samples were loaded on previously outgassed 99.98% single Re filaments using 2.5µl of a Ta$_2$O$_5$-H$_3$PO$_4$-HF activator solution. The analyses ran in a dynamic multi-collection mode at analyzing intensities ≥ 1V for 88Sr and temperatures between 1300–1350 °C. Runs of the standard reference material SRM 987 returned an average $^{87}$Sr/$^{86}$Sr ratio of 0.710238 ± 0.00002 (n = 10, 2σ), which is slightly offset from the mean SRM 987 value of 0.710245 published by [86]. The measured $^{87}$Sr/$^{86}$Sr values of the samples were corrected accordingly. The within-run precisions (2SE) of the individual runs were up to 0.00006, but generally ≤ 0.00003 (Table 1). Procedural blanks were in the order of 20 pg to 60 pg of strontium, with blank $^{87}$Sr/$^{86}$Sr signatures between 0.7095 and 0.7112. However, no blank correction was conducted as the blank contribution is considered insignificant compared to the overall sample Sr of typically ≥200ng.

## Results

### Kalvehavegård and Sølager data

The strontium isotope results from the Bronze Age sites of Kalvehavegård and Sølager/Store Karlsminde are presented in Table 1.

At Kalvehavegård, the $^{87}$Sr/$^{86}$Sr values range from 0.70929 to 0.71017, while values from Sølager range from 0.70974 to 0.71577 (Table 1). Data are displayed in Fig 4. Multi-tooth enamel analyses from individuals at Kalvehavegård lie close to each other and thus indicate limited individual mobility from early childhood to adolescence. On closer inspection though, except for the individual from grave 77, we note a systematic lowering of $^{87}$Sr/$^{86}$Sr values from the first (M1) molar to later mineralized molars (M2, M3). The reason for this cannot be established at this stage, but further, more detailed studies may eventually be helpful in elucidating whether such a chronological trend is pertinent to other individuals, also from different sites.

At Sølager, while most individuals exhibit a relatively tight range in $^{87}$Sr/$^{86}$Sr values, a few individuals (graves 4, 13,14,17,28, 31a) are characterized by elevated strontium isotope signatures. These elevated values are recorded in both enamel and pars petrosa samples.

### The $^{87}$Sr/$^{86}$Sr database of archaeological human remains

We compiled all published $^{87}$Sr/$^{86}$Sr values, complemented with strontium concentrations [Sr] where available, of individual human tooth enamel and pars petrosa samples and combined them with the new $^{87}$Sr/$^{86}$Sr values presented herein. This entire dataset is listed in S1 Table of the supplementary information. Enamel and pars petrosa samples that were prepared and measured by the first and last author in collaboration with the Danish Center for Isotope Geology, Department of Geoscience and Natural Resource Management, University of Copenhagen, are highlighted in grey color in S1 Table.

Based on the crude categorization into archaeological periods, a chronological order of the data set is represented in Fig 5. We have applied a "Median Average Deviation; MAD" statistical approach to the data set, with the aim of identifying outliers in the skewed data distribution presented in the histogram of Fig 6. We chose this approach because the MAD statistical method appreciates robustness of the major tendency of the data distribution, which is characterized by the median value. In our dataset, a large number of strontium isotope signatures of human remains are grouped around the median, and we suggest that they most likely best characterize the true range of bioavailable strontium isotope signatures

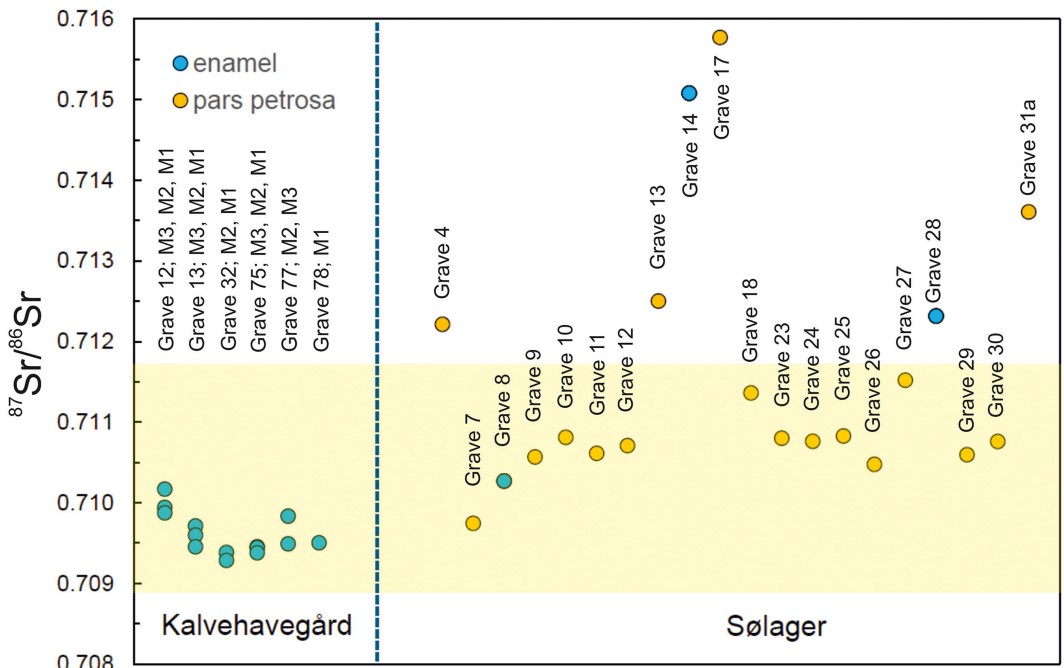

**Fig 4. Diagram of strontium isotope data of individual humans from Kalvehavegård and Sølager.** Data from Kalvehavegård show little variation in their $^{87}Sr/^{86}Sr$ signatures. Multi-tooth enamel analyses from individuals also depict only small differences in their strontium isotope signatures, suggesting the absence of mobility in childhood through adolescence periods of the respective individuals. In contrast, the individuals from Sølager yield a much larger variation in their $^{87}Sr/^{86}Sr$ values. This larger variation, and in particular individuals with unusually high strontium isotope ratios compared to the human-based bioavailable strontium isotope range calculated in this study (marked by a transparent yellow band across the diagram) in graves 4, 13,14, 17, 28 and 31a, suggest potential mobility.

in the past within present-day Denmark (Bornholm excluded). The MAD method has been previously proposed, together with the interquartile range (IQR) approach, to be the most appropriate method to designate migrants in human archaeological population samples in a worldwide survey dataset of oxygen isotopes in human tooth enamel and bone bioapatite samples ([87]). The median value "M" is at $^{87}Sr/^{86}Sr$ = 0.71032, and outliers are statistically defined from the distribution of absolute deviations of the sample values from the median value (MAD) as follows:

$MAD = b * M_i * (|x_i − M_j(x_j)|)$; [88]

where the $x_j$ is the n original observations, $M_i$ is the median of the series, and b = 1.4826, a constant linked to the assumption of normality of the data, disregarding the abnormality induced by outliers [89]. The outlier criterion then is defined as:

$M − 2 * MAD < x_i > M + 2 * MAD$

In the case of the concrete data set presented herein, where the median $^{87}Sr/^{86}Sr$ value is 0.71032, outliers are defined as:

$$0.70862 > x_{outlier} > 0.71194$$

The use of a multiplying factor of 2 for MAD, instead of a factor of 2.5 (described by [89] as a "poorly conservative" approach), conveniently allows for the capture of outliers in our dataset. The outliers are contained in the flat tail of the data distribution and in the slightly asymmetric right part of the main data frequency distribution in the histogram of Fig 6, where data are also characterized by slightly elevated $^{87}Sr/^{86}Sr$ data. The effectiveness of outlier removal can be seen in the resulting very low values for the skewness and kurtosis parameters of the data distribution after outlier removal as

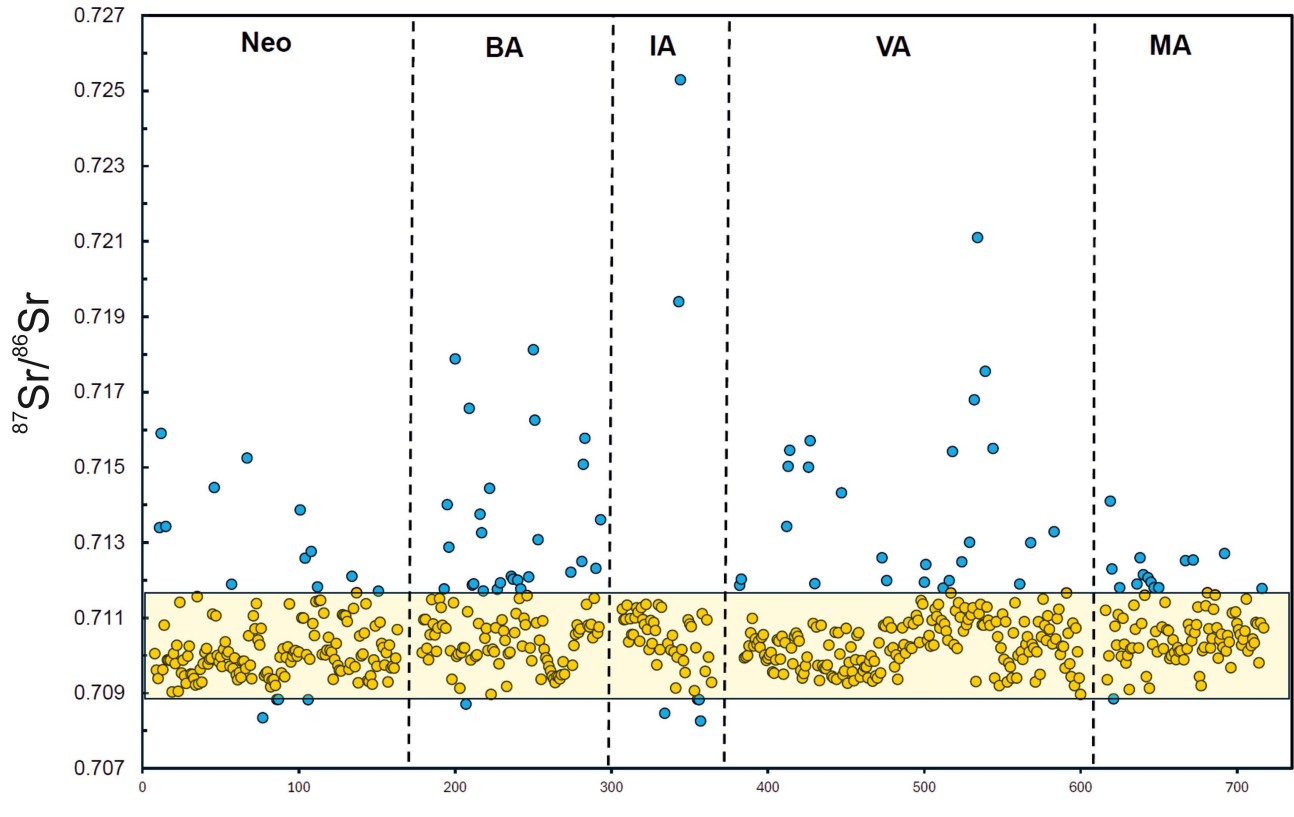

**Fig 5. $^{87}$Sr/$^{86}$Sr values of human individuals categorized by time periods.** Diagram depicting the time-categorized $^{87}$Sr/$^{86}$Sr values of tooth enamel and pars petrosa samples of individual humans. The yellow-filled symbols depict values that define the outlier-free human-based bioavailable strontium isotope range. The blue-filled symbols mark samples which plot above and beneath the upper and lower range limits, respectively.

listed in Table 2. Concretely, from the 628 human $^{86}$Sr/$^{86}$Sr analyses, 64 analyses (i.e., 11.3%) are classified as outliers in that way (marked with blue symbols in Fig 5). Additional conventional descriptive statistical parameters of the MAD outlier-free data set are also listed in Table 2:

Fig 7 presents the corresponding histogram. The outlier-free range of $^{87}$Sr/$^{86}$Sr = 0.70891–0.71171 (mean = 0.71031 ± 0.00140; n = 564, 2σ) is interpreted as the human-based bioavailable Sr isotope range for Denmark (excluding Bornholm). This range is indicated on the timeline diagram in Fig 5 by a yellow semi-transparent rectangle for reference. Values below 0.70891 and above 0.71171, representing the lower and upper limits of the range, respectively, comprise 90 data points (highlighted in blue-filled symbols).

## Discussion

### Interpreting the human-based bioavailable strontium isotope range

From an overall perspective, the human-based strontium isotope range represents the bioavailable signature of ~86% of all 628 individuals of the compiled dataset. We therefore consider this range as very representative of Denmark's past bioavailable strontium isotope composition. When compared with the previously established surface-water based

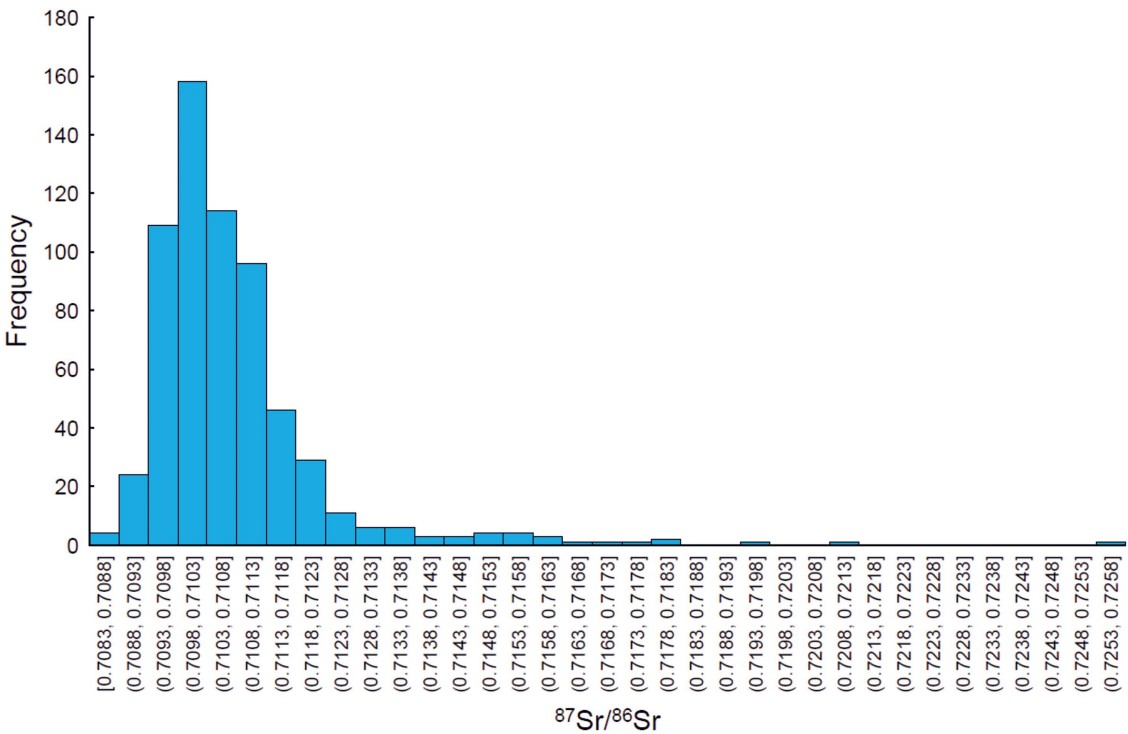

**Fig 6. Histogram of all 628 human strontium isotope data.** Histogram comprising the entire data set of currently available and new strontium isotope signatures presented herein. The enhanced skewness of the data distribution attests to a significant proportion of outliers which do not follow a normal Gaussian-type statistical distribution.

**Table 2. Descriptive statistics parameters of outlier-free human-based dataset.**

| | |
|---|---|
| Mean | 0.71031 |
| Standard Error | 0.00003 |
| Median | 0.71023 |
| Standard Deviation | 0.00070 |
| Kurtosis | −0.58 |
| Skewness | 0.22 |
| Range | 0.00322 |
| Minimum | 0.70871 |
| Maximum | 0.71193 |
| Count | 564 |

baseline([17], $^{87}Sr/^{86}Sr = 0.7081–0.7111$) for present-day Denmark (Bornholm excluded), the human-derived isotope range presented herein ($^{87}Sr/^{86}Sr = 0.7089–0.7117$) is slightly shifted toward elevated $^{87}Sr/^{86}Sr$ signatures. This subtle difference may reflect the somewhat elevated strontium isotope ratios of plants (and thus plant-based foodstuffs) consumed by the individuals. These plant signatures in turn likely mirror the progressive leaching of glaciogenic soils containing natural carbonate components over time as proposed by [16]. Since the last glacial retreat in this region, such acid leaching has likely produced increasingly radiogenic bioavailable signatures in soils, compared to the natural carbonate-buffered voluminous surface and groundwater sources. For further details on the newly established human-based bioavailable

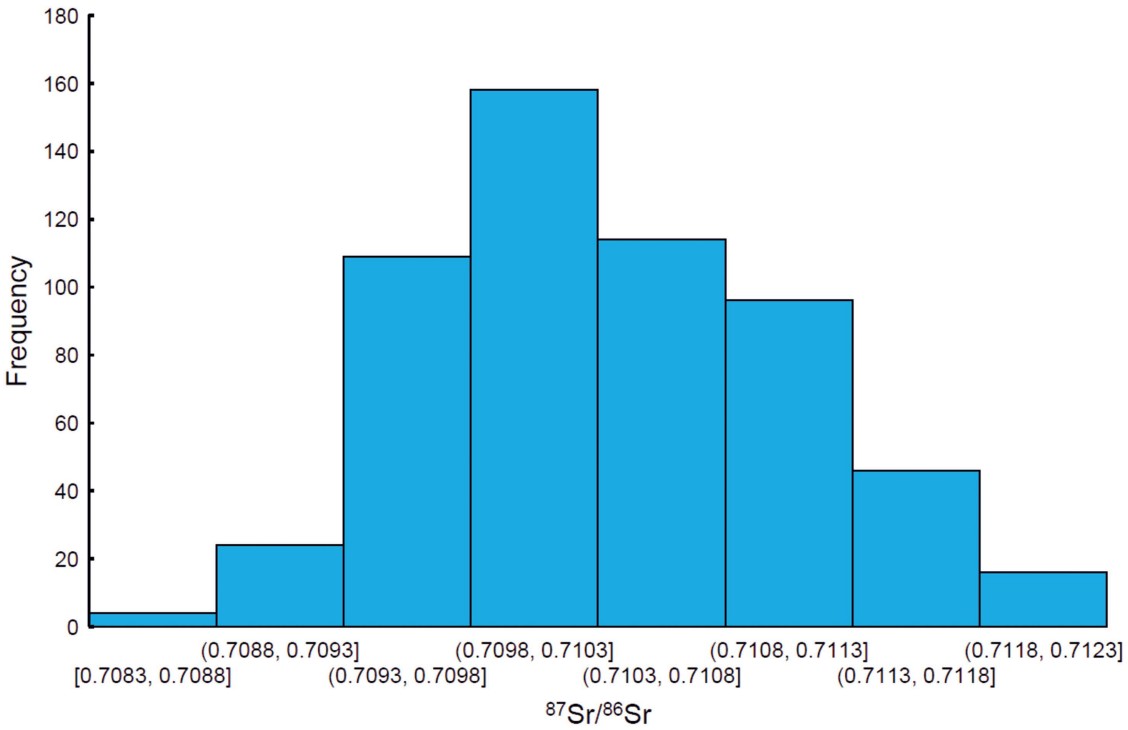

**Fig 7. Histogram of the outlier-free data set.** Histogram depicting the distribution of the individual human strontium isotope signatures after application of the MAD criteria to define statistical outliers (descriptive statistical parameters listed in Table 2).

strontium isotope range, and its comparison with previously published proxy-based baselines for present-day Denmark, see the S1 Appendix in the Supplementary Information with some additional references ([90–102]) that are not directly cited in the main text herein.

### Interpreting the strontium isotope results of the non-elite Bronze Age individuals

Within the Nordic Bronze Age, renowned for its rich elite burials, non-elite individuals have long remained rather "invisible" in the archaeological record (e.g., [8]). While some of the best-preserved elite Bronze Age burials in Europe have been uncovered within present-day Denmark, the non-elite segment of the population—which is assumed to have formed the majority, with some researchers suggesting as much as 80% [103]—has remained largely uninvestigated. Bergerbrant and coworkers [8] highlight how this significant proportion of non-elite burials (including those of children) had previously been "mysteriously" missing from the archaeological Nordic Bronze Age record. Their study investigated 65 burial samples from southern Sweden that were originally contextually dated to the Late Neolithic and Early Bronze Age. The authors conducted a series of analyses, including osteological assessments, strontium isotope measurements, ancient DNA (aDNA) studies, and radiocarbon dating. Because the samples were chosen based on the preservation of teeth suitable for aDNA analysis, the material encompassed a variety of grave types. Surprisingly, their radiocarbon dating revealed that many individuals previously thought to belong to the Late Neolithic actually dated to the Early Nordic Bronze Age. This discovery helped to partially explain the apparent absence of non-elite burials from the period and, for the first time within the Nordic Bronze Age context, allowed a comparison of individual mobility patterns between elite and non-elite individuals of the Early Nordic Bronze Age [8]. However, to date, no comparable study has been carried out within the region of present-day Denmark. Hence, our investigation set out to fill this gap and to, for the first time, shed light on the

sociodynamics of non-elite individuals dating to the Nordic Bronze Age unearthed within present-day Denmark. Further-more, we also extend the temporal framework (with respect to the study in Sweden which investigated individuals from the Early Nordic Bronze Age) by including a cemetery of non-elite burials from the Late Nordic Bronze Age, as exemplified by the site of Sølager.

The Bronze Age burial site of Kalvehavegård offers a distinctive combination of funerary practices. On one side, the central grave contained a richly furnished cremation burial in an oak-coffin, dated to Period II of the Early Bronze Age (1500–1300 BCE), and accompanied by elite grave goods, including gold ornaments and fittings interpreted as part of a folding chair. This central grave contained the cremated human remains of two individuals; however, no suitable samples from these were identified from which strontium isotope analyses could be conducted. On the other side, just outside the edge of the barrow's northern rim, a flat field cemetery was also found. These series of inhumation graves, representing men, women, and children with little or no grave goods, clearly reflect a non-elite segment of society. Radiocarbon analyses conducted on some of these individuals revealed that they were slightly older than the central grave, dating mostly to the Early Nordic Bronze Age period I (1700–1500 BCE). Interestingly, the individuals in these graves were buried following the Late Neolithic tradition [9], simi-larly to the findings of Bergerbrant and coworkers [8] in southern Sweden. The coexistence of different grave traditions within the same burial ground (i.e., an elite cremation as the central grave, and non-elite inhumation graves at the outer rim of the barrow), illustrates the intersection of emerging cremation practices with older traditions. This provides a rare insight into social differentiation in Early Nordic Bronze Age communities. In our present investigation, these inhumation graves of the flat field cemetery represent the early phase of non-elite individuals from the Nordic Bronze Age.

Before presenting the specific mobility interpretations, it is important to clarify how the terms *"local"* and *"non-local"* are used in the following discussion, as these classifications depend on the chosen baseline or reference dataset and its geographic scope.

When referring to the surface-water baseline—which defines a country-wide strontium isotope range for present-day Denmark (excluding Bornholm)—an individual classified as *non-local* is interpreted as originating outside present-day Denmark (excluding Bornholm). Conversely, individuals whose values fall within this range are considered *local*, meaning they likely originated either within present-day Denmark or in another region with an overlapping strontium isotope range.

When using the fauna-based baseline, which distinguishes between slightly different ranges for western Denmark (Jut-land) and eastern Denmark (including Funen and Zealand but excluding Bornholm), the terms *local* and *non-local* apply specifically to these sub-regions.

Finally, when interpreting the results in relation to the human-based bioavailable range defined in this study, *local* refers to origins within present-day Denmark excluding the island of Bornholm, while *non-local* indicates values falling outside this human-derived range.

The strontium isotope values yielded by the individuals from Kalvehavegård range from $^{87}Sr/^{86}Sr = 0.70929$ to $0.71017$. When comparing these with the proposed baselines for present-day Denmark [17,37] they all fall within these baselines. Furthermore, they also fall within the baseline proposed for the eastern part of Denmark in [37], which includes the island of Funen where the site of Kalvehavegård is located. Considering these two baselines, the strontium isotope results of the individuals from Kalvehavegård seem to be quite local to the area where they were buried. When comparing the strontium isotope values yielded by the Kalvehagård individuals to the human-based bioavailable strontium isotope range calculated herein (marked as a yellow band in Fig 4), all the six individuals investigated herein also fall within this range, which provides an additional reference comparison that points to their local origin. However, given the site's coastal location and the presence of marine shells among the grave goods, we cannot exclude that the individuals' diets included a proportion of marine foods. If so, the strontium isotopic com-position of their tooth enamel may have been slightly skewed toward marine signatures of $^{87}Sr/^{86}Sr = 0.7092$ [104]. However, as all the strontium isotopic signatures of the individuals fall considerably below the upper range limit, it suggests that their local signatures are rather solid.

Interestingly, we were able to obtain strontium isotope values from first, second, and third molars, for three individuals (graves 12, 13, 75; Fig 4), thus covering a developmental timeline from early childhood to adolescence. Their isotopic compositions are highly similar, suggesting that these individuals resided within the same area throughout their early lives. Hence, it seems that within the Kalvehavegård site, mobility was rather limited.

The second site we investigated is the Late Nordic Bronze Age (1000–800 BCE) cemetery at Sølager, which comprised 31 stone-setting burials, most containing ceramic urns with the cremated remains of single individuals [78]. The graves were modest, with few artefacts, indicating a community of limited wealth. Important to our investigation is that these non-elite burials represent the later period of the Nordic Bronze Age as compared to the individuals buried at Kalvehavegård. The central urn grave was covered by a slightly elevated burial mound while the remaining urns were buried in the periphery of the burial mound (see Fig. 4). The arrangement of stone-setting burials most likely points to some kind of a community group [78]. Notably, two female burials contained fishhooks—an unusual form of grave good that may reflect a subsistence strategy involving fishing.

The strontium isotope values yielded by the individuals from Sølager range from $^{87}Sr/^{86}Sr = 0.70974$ to $0.71577$ and thus have a much larger spreading in their isotopic values compared to the range observed in the site of Kalvehavegård. When comparing the strontium isotope values yielded by the individuals from Sølager with the proposed baselines for present-day Denmark [17,37] some of the individuals fall outside these baselines. When compared to the country-wide surface water baseline [17], eight of the 20 individuals investigated fall outside the baseline (graves 4,13, 14, 17, 18, 27, 28 and 31a), suggesting mobility and non-local origin to the region of present-day Denmark. Yet, in relation to the baseline proposed for the eastern part of Denmark in [37], which includes the island of Zealand where the site of Sølager is located, two additional individuals appear to be of non-local origin (graves 10 and 25). Considering these two baselines, the strontium isotope results of the individuals from Sølager point to a considerable high degree of mobility within this group of non-elite individuals. When comparing the strontium isotope values yielded by the Sølager individuals to the human-based bioavailable strontium isotope range calculated herein (marked as a yellow band in Fig 4) only six yielded signatures outside the range. So, while applying the previously published baselines, ten to eight individuals point to non-local origins, yet only six seem to be non-local when compared to the human-based bioavailable strontium isotope range calculated herein. Interestingly, these six non-local individuals define an expanded range in their $^{87}Sr/^{86}Sr$ values, suggesting migration from different areas abroad. Individuals from graves 4, 13 and 28 (Fig 4) have similar values, possibly suggesting a shared, geographically non-local origin outside present-day Denmark.

The three remaining non-local individuals with the highest $^{87}Sr/^{86}Sr$ ratios (graves 14, 17, and 31a; Fig 4) likely came from one or more regions outside present-day Denmark that are different from the areas constrained by the other three non-local individuals. If we look at the two females with fishhooks (graves 10 and 24), they both yielded local signatures when compared to the surface-water baseline and when compared to the human-based bioavailable strontium isotope range calculated herein (Fig 4). However, the female in grave 10 falls just outside the baseline proposed for the eastern part of Denmark in [37]. Despite the differences in the number of identified non-locals depending on which reference material is applied, they all point to mobility of at least six of the 20 individuals investigated.

Whether the Søhale cremated remains reflect a marine dietary contribution—hinted at by the associated fish bones—remains uncertain. As Snoek and coworkers [105] note, the high temperatures of cremation alter bone carbon and oxygen isotopes, limiting dietary insights. While the high temperatures (up to 1000°C and above) destroy most organic matter such that also the extraction of reliable DNA from intensively burned human remains is extremely difficult, strontium concentrations and isotopes are preserved during cremation and can be used to assess the geographical origin of unidentified fire-affected individuals (e.g., [76,106]). If marine resources did influence the Sølager community's diet, the measured $^{87}Sr/^{86}Sr$ values (Table 1, Fig 4) represent minimum estimates for their terrestrial-based diet, because modern seawater—and thus marine foods—has a characteristic ratio of 0.7092 [104], which would lower terrestrial biosignatures. Consequently, the six non-local signatures at Sølager remain robustly non-local even if marine foods were consumed.

A particularity of the group of non-locals whose similar strontium isotopic composition might suggest a shared, geographically non-local origin outside present-day Denmark (see above) is that grave 28 contains the remains of a small child (Table 1; Figs 3,4). This sample shows a non-local $^{87}Sr/^{86}Sr$ ratio in a deciduous tooth. Because the child was likely breastfeeding, this signal could reflect the mother's isotopic signature rather than the child's own diet. Another interesting aspect is that the child's signature is similar to the signatures of two other individuals (graves 4 and 13; Table 1; Fig 4) which could suggest a common geographical origin of these three individuals.

Taken together, these findings demonstrate that the complexity of mobility previously identified in elite burials is also reflected among non-elite individuals. In other words, mobility from regions outside present-day Denmark was not restricted to the elite, as evidenced by the presence of non-locals at Sølager.

## Kalvehavegård and Sølager in the framework of the Nordic region

In order to compare our findings, we draw on the previously mentioned study from southern Sweden, which a few years ago carried out the first investigation of non-elite burials from the Early Nordic Bronze Age [8]. That study highlighted the problem that our current knowledge of the Nordic Bronze Age is largely based on elite burials, implying that a substantial proportion of the population (i.e., the non-elite one) remains largely invisible in the archaeological record. Our investigation therefore provides valuable information on this 'missing' segment of the Nordic Bronze Age population, while also offering a rare opportunity for direct comparison with the southern Swedish material [8]. It also expands the timeframe by incorporating individuals dating to the Late Nordic Bronze Age from the site of Sølager. The Swedish study [8] states that "strontium isotope analyses reveal that individuals seem to be mobile regardless of their wealth status and burial rituals. It suggests a society where workers and perhaps even non-free labourers were mobile, not only the elite segment". Our findings from present-day Denmark reveal a similar pattern, at least during the Late Nordic Bronze Age period, to those previously documented in southern Sweden, during Early Nordic Bronze Age. In essence, our findings suggest that the mobility of non-elite individuals was not confined to southern Sweden but represented a broader phenomenon across the Nordic region during this entire Nordic Bronze Age. This suggests that the role of non-elite groups in shaping Bronze Age society was probably much greater than previously anticipated. In other words, while the non-elite individuals may not have left substantial archaeological cultural remains in the form of grave goods, the social impact of non-local, non-elite individuals on Nordic Bronze Age society was likely still present. Future theoretical frameworks could be developed on the basis of these new findings in order to investigate how such non-local, non-elite individuals influenced and contributed to the societies they became part of.

## Mobility through time

We think that the potential of this new database of published and herein newly presented strontium isotope values from human remains, provides a strong framework for future comparisons and further in-detailed studies of this large data. Therefore, we also made an attempt to calculate the proportions of individuals that fall outside the human-based bio-available strontium isotope range calculated herein (blue filled symbols in Fig 5) for each of the periods. These shares are as follows: ~10% (16 out of 156) in the Neolithic, ~25% (28 out of 112) in the Bronze Age, ~12% (6 out of 51) in the Iron Age, ~12% (25 out of 211) in the Viking Age, and ~15% (15 out of 98) in the Medieval period. Assuming that these individuals which fall outside this range represent mobility, and in light that these shares are conformable with the mobility range of 5–30% expected even in past sedentary societies, as proposed by a large group of leading archaeologists at the Summit for Social Archaeology and Climate Change in Kiel (Germany) at the EAA conference in 2021 (page 10 of the statement; contact persons Prof. Peter F. Biehl, USA, and Prof. Johannes Müller, Germany), it seems reasonable to propose that the human-based bioavailable strontium isotope range calculated herein is representative of the characteristics of the bioavailable strontium isotope baseline of the past. In this light, and based on the current human data available

in our database, it looks like mobility within the region of present-day Denmark was most pronounced during the Bronze Age, compared to the other periods.

## Conclusions

As strontium isotope data from human remains continue to grow and research questions on past mobility expand, comprehensive datasets become increasingly important as reference material for interpreting $^{87}Sr/^{86}Sr$ values. At the same time, the accumulation of analyses highlights periods and social groups that remain underrepresented. One such gap concerns non-elite individuals from the Nordic Bronze Age in present-day Denmark.

Our study is the first to present strontium isotope analyses of non-elite human remains from this region and period. We conducted 34 analyses on individuals from two sites: six inhumations from the Early Nordic Bronze Age site of Kalvehavegård and twenty cremated individuals from the Late Nordic Bronze Age site of Sølager. To contextualize these results, we assembled a comprehensive dataset of all published human $^{87}Sr/^{86}Sr$ values from Denmark to date (n=513) and combined them with new generated data and previously unpublished data (n=115, including the 34 Bronze Age samples). The resulting dataset comprises 628 strontium isotope values from enamel and pars petrosa samples from individuals spanning the Mesolithic to the Medieval period.

Using a conservative approach based on the median absolute deviation (MAD) to identify outliers (n=564), we define a human-based bioavailable range of $^{87}Sr/^{86}Sr$=0.7089–0.7117 (mean±2σ: 0.7103±0.0014). We compare this human-based range to previously proposed baselines for Denmark and interpret the non-elite individuals' values against both reference frameworks.

Regardless of which baseline is applied, all individuals from Kalvehavegård fall within the local range, indicating very limited mobility at this Early Bronze Age site. In contrast, between 8 and 10 of the 20 individuals from Sølager fall outside previously proposed baselines for Denmark, and six fall outside the new human-based range. Together, these results clearly indicate mobility among non-elite individuals at Sølager, with at least six people originating from beyond present-day Denmark (excluding Bornholm). These findings demonstrate that mobility during the Nordic Bronze Age was not confined to elite segments of society. When combined with findings from Early Nordic Bronze Age southern Sweden, our results show that non-elite mobility formed an integral part of wider regional socio-dynamics and cultural exchange processes that influenced the development of the Nordic Bronze Age communities.

Finally, the human-based strontium isotope dataset and the derived corresponding range—representing approximately 86% of all archaeological human strontium isotope values published to date for present-day Denmark (excluding Bornholm)—provide a robust diachronic reference framework for future studies. This dataset not only strengthens the interpretation of new $^{87}Sr/^{86}Sr$ analyses but also highlights remaining gaps that future research can address.

## Supporting information

**S1 Table. Published and new strontium isotope signatures of ancient human enamel and pars petrosa samples from Denmark (Bornholm excluded).**
(XLSX)

**S1 Appendix. Significance of human-based strontium isotope range and its comparison with modern proxy-based baselines.** This section describes the significance and relevance of the human-based strontium isotope range presented herein and compares it with other environmental proxy-based baselines that have been published for Denmark.
(DOCX)

**S1 Fig. Temporal distribution diagram focusing on data of human-based strontium isotopes range-defining local individuals.**
(TIF)

**S2 Fig. Temporal distribution diagram comparing the human-based strontium isotope range with the surface water-based baseline.**
(TIF)

**S3 Fig. Temporal distribution diagram comparing the human-based strontium isotope range with the soil leachate-based baseline.**
(TIF)

**S4 Fig. Temporal distribution diagram comparing the human-based strontium isotope range with the plant-based baseline.**
(TIF)

## Acknowledgments

We thank Cristina Jensen for help with sample preparations and ion chromatographic separations in the isotope laboratories of the Danish Center for Isotope Geology, University of Copenhagen. We also thank Marie Louise Jørkøv for assistance in the Anthropological Laboratory, University of Copenhagen, when choosing the samples from the Sølager site. Finally, we thank Samantha S. Reiter for her assistance in securing the transport to the laboratory of some of the samples from Kalvehavegård.

No permits were required for the described study, which complied with all relevant regulations.

## Author contributions

**Conceptualization:** Karin Margarita Frei.

**Data curation:** Karin Margarita Frei, Robert Frei.

**Formal analysis:** Karin Margarita Frei, Robert Frei.

**Funding acquisition:** Karin Margarita Frei.

**Investigation:** Karin Margarita Frei, Robert Frei.

**Methodology:** Karin Margarita Frei, Robert Frei.

**Project administration:** Karin Margarita Frei.

**Resources:** Karin Margarita Frei, Malene Refshauge Beck, Pernille Pantmann, Niels Algreen Møller, Morten Søvsø.

**Supervision:** Karin Margarita Frei.

**Writing – original draft:** Karin Margarita Frei, Robert Frei.

**Writing – review & editing:** Karin Margarita Frei, Malene Refshauge Beck, Pernille Pantmann, Niels Algreen Møller, Morten Søvsø, Robert Frei.

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
