## [Decision Letter · Decision Letter 0]

12 Nov 2025

Dear Dr. Frei,

Thank you for submitting your manuscript to PLOS ONE. After careful consideration, we feel that it has merit but does not fully meet PLOS ONE’s publication criteria as it currently stands. Therefore, we invite you to submit a revised version of the manuscript that addresses the points raised during the review process.

The manuscript by Frei et al. presents a rich database aimed at creating a baseline specific to Denmark for the 87/86 strontium isotope ratio.

Both reviewers welcomed the manuscript but also raised some issues and topics for discussion that need to be addressed by the authors in order to make the manuscript publishable in Plos One.

In particular, reviewer 2 raised some doubts about the basis of the analysis itself. As far as I am concerned, I believe that the manuscript  should take into account the reviewers' criticisms, considering them as constructive contributions (even beyond the form). The criticisms should be integrated into the text, also showing possible alternative scenarios with a less assertive and more critical approach. Currently, the manuscript has a dual hybrid dimension, the first being a review paper (without the necessary insights), and the second one being a definition of the baseline (without an effective critical approach).

Beyond these general considerations, I must add a series of notes:

1. The database in Table S1 must also include a column indicating whether the individuals were cremated or inhumed.

2. The database contains values with four or five decimal places: where possible, it would be advisable to standardise the values as much as possible to the fifth decimal place.

3. I have carried out a quick analysis of the data in Table 6: as can be seen in the attached PDF file, 40.4% of the isotopic ratio values are present in at least two individuals, up to a maximum of six individuals, all with a value of 0.7102, and six with a value of 0.7101. Twenty-three per cent of the duplicates have isotope ratio values to four decimal places, but the remaining 77% are values to five decimal places. Although a certain percentage of duplicate data may be “reasonable” in an analysis of this type, I feel that these percentages are excessive and I invite the authors to further check Table S1. Finally, I think that this situation, if confirmed, should be discussed in the article and justified.

4. In my opinion, the use of MAD should be better justified with more literature. Furthermore, the choice to use a multiplier of 2 to define the range of “local” individuals should be better justified (also because the bibliography cited by the authors suggests a value of 2.5). I have taken the liberty of attaching a graph showing the different ranges resulting from different multiplier values. Interestingly, the value of 2.5 seems to be close to the value of Tukey's method, which is the most widely used in the literature for defining outliers.

We look forward to receiving your revised manuscript.

Kind regards,

Luca Bondioli, PH.D.

Academic Editor

PLOS ONE

Journal Requirements:

2. In your manuscript, please provide additional information regarding the specimens used in your study. Ensure that you have reported human remain specimen numbers and complete repository information, including museum name and geographic location.

For more information on PLOS One's requirements for paleontology and archeology research, see https://journals.plos.org/plosone/s/submission-guidelines#loc-paleontology-and-archaeology-research .

This study was made possible through the funding to KMF provided by the Carlsberg Foundation “Semper Ardens” advance research grant CF18-0005 for which we are very grateful.

https://www.carlsbergfondet.dk/en

4. We note that Figure 1 in your submission contain map images which may be copyrighted. All PLOS content is published under the Creative Commons Attribution License (CC BY 4.0), which means that the manuscript, images, and Supporting Information files will be freely available online, and any third party is permitted to access, download, copy, distribute, and use these materials in any way, even commercially, with proper attribution. For these reasons, we cannot publish previously copyrighted maps or satellite images created using proprietary data, such as Google software (Google Maps, Street View, and Earth). For more information, see our copyright guidelines: http://journals.plos.org/plosone/s/licenses-and-copyright.

Reviewers' comments:

Reviewer's Responses to Questions

**Comments to the Author**

1. Is the manuscript technically sound, and do the data support the conclusions?

Reviewer #1: Yes

Reviewer #2: Partly

2. Has the statistical analysis been performed appropriately and rigorously?

Reviewer #1: N/A

Reviewer #2: No

3. Have the authors made all data underlying the findings in their manuscript fully available?

Reviewer #1: Yes

Reviewer #2: Yes

4. Is the manuscript presented in an intelligible fashion and written in standard English?

Reviewer #1: Yes

Reviewer #2: Yes

Reviewer #1: The paper of Frei et al reviews the human Sr isotope data from Denmark and uses this dataset as a way to detect local and non-local individuals among two archaeological sites. I have only minor comments that follow.

I do believe lines from 45 to 55 should be moved at the end of the introduction.

The introduction is a bit weirdly structured: from the title and the abstract a reader expects to find a rationale behind the use of human samples to reconstruct local bioavailable Sr and then the application of this to a case study (non-elite individuals.). Overall, I think the authors should adjust the introduction following the abstract flow.

Figure 1: I think a (simplified) geological map would be important to show.

Line 128: “we compile 513 previously published strontium isotope” please indicate here from how many sites.

Table 2: please use a consistent and reasonable number of digits.

Table S1: why for some samples the 2SE is not reported?

Line 382: the result section starts from the case study and then jump to the Denmark baseline definition. I describing the baseline should come before.

Line 427: the authors used the entire dataset for the MAD approach. This means that the ‘baseline’ they calculate and refer to encompasses all of Denmark. Consequently, individuals identified as ‘non-local’ are likely those originating from outside the current boundaries of Denmark. While this approach is valid, the authors should clarify this methodological choice and provide a rationale for it. In contrast, my preferred approach would have been to calculate separate MAD values for each area or sub-region within Denmark. In addition, I think it would be important to clarify, maybe in the method section, what the authors mean by ‘non-local’.

Line 427-437: I think this part pertains the method section

Reviewer #2: General Assessment

This manuscript presents an ambitious and valuable dataset, compiling 628 human 87Sr/86Sr values from archaeological sites across Denmark. This database, in itself, is a significant contribution and will serve as an important reference for future work in the region.

However, the manuscript's central methodological premise and its primary conclusion—that a single, nationwide "baseline" (0.7089–0.7117) can be derived from this national dataset using simple statistics and then be applied to provenance individuals—is fundamentally flawed.

The authors correctly criticize the limitations of environmental isoscapes (e.g., those based on modern water or soil), but they fail to demonstrate that their proposed alternative is a more suitable or accurate tool for provenancing. In fact, the proposed method ignores the single most important factor in strontium-based mobility studies: spatial variability.

Major Points for Revision

1. The Methodological Fallacy of a Single National Baseline

The core flaw of this paper is the assumption that a single 87Sr/86Sr range can represent "local" for the entire nation of Denmark. The authors' own sources and the wider literature (e.g., Frei & Frei, 2011) show that bioavailable strontium is spatially variable across Denmark due to its complex glacial geology.

The authors' approach conflates two different concepts: "local" (an individual originating from the specific geological catchment of the burial site) and "Danish" (an individual originating from anywhere within the modern political borders of Denmark).

By "flattening" all of Denmark's diverse geological signatures into a single statistical range, the authors' baseline becomes (a) too broad to detect inter-regional mobility and (b) too narrow to correctly identify locals from high- or low-Sr regions of Denmark. For example, an individual moving from a high-Sr region (e.g., 0.7115) to a low-Sr region (e.g., 0.7090) would be classified as "local" in both places, rendering the baseline ineffective for tracking mobility within Denmark.

2. Misapplication of Statistical Methods

The statistical method chosen (Median Absolute Deviation, or MAD, to clean the data, followed by 2 sigma to define the range) is a valid approach for defining a local baseline at a single site, where it is reasonable to assume the majority of individuals are local to that site's specific geology.

However, applying this method to a composite national dataset drawn from hundreds of different sites with different geological backgrounds is statistically inappropriate. The resulting range (0.7089–0.7117) is not a "baseline" for mobility studies; it is merely a statistical summary of the central 95% of all bioavailable Sr values in Denmark.

This method statistically defines individuals from the highest- and lowest-Sr regions of Denmark as "non-local," which is a critical error. This has direct implications for the case study: the six "non-local" individuals from Sølager may not be from outside Denmark; they may simply be from a part of Denmark with a naturally high-Sr signature that was statistically trimmed from the "local" range by the MAD analysis.

3. Failure to Demonstrate Suitability (A Path Forward)

The authors criticize environmental isoscapes but fail to provide any evidence that their method is superior. To validate their claims, they must directly compare their "simple baseline" against a spatially-explicit model.

Given that the 628 data points in the database are geolocated (as implied by Figure 1), the authors are in a prime position to do this. The critical revision required for this manuscript is as follows:

• The authors must use their 628 human data points to generate a human-derived 87Sr/86Sr isoscape for Denmark using appropriate spatial statistics (e.g., kriging).

• They must then compare the predictive power of this new isoscape against their "simple baseline."

• This re-analysis would allow them to re-evaluate the Sølager and Kalvehavegård individuals against a spatially-relevant local value (as predicted by their new human isoscape), rather than an arbitrary national average.

Without this spatial analysis, the paper's central argument is unsubstantiated, and its conclusions regarding non-elite mobility are unreliable.

Conclusion

The dataset presented is of great value to the archaeological community. However, the methodology used to interpret this data is flawed and ignores the foundational principles of strontium provenance. The "simple baseline" approach is inappropriate for a nationwide study.

I recommend Major Revisions. The paper cannot be accepted until the authors address this fundamental methodological flaw, preferably by conducting the spatial analysis suggested above and re-evaluating their conclusions accordingly.

**Do you want your identity to be public for this peer review?** For information about this choice, including consent withdrawal, please see our Privacy Policy

Reviewer #1: No

Reviewer #2: No

---

## [Author Response · Author response to Decision Letter 1]

11 Dec 2025

Response to reviewers

PONE-D-25-53013

Original title: A new human-based strontium isotope baseline for Denmark and its application to Bronze Age non-elite mobility

New revised title: Bronze Age non-elite mobility in Denmark examined through a new human-based bioavailable strontium isotope range

We sincerely appreciate the detailed and constructive comments provided by both reviewers and the academic editor, Dr. Luca Bondioli. We are pleased that the reviewers agree in that the compiled and new supplemented human dataset are a valuable resource for future archaeological provenance studies in Denmark. While reviewer 1# has only minor comments, we are pleased to also read the conclusion of reviewer #2 who writes that “the dataset presented is of great value to the archaeological community”. We are therefore grateful that both reviewers see the value of this study and contribution to the research community. At the same time, we acknowledge that several aspects of our original manuscript were not presented with sufficient clarity, which contributed to some misunderstandings. We have now clarified these points in the revised version.

Below, we outline the revisions we have made in response to the reviewers’ thoughtful comments. We address each point individually in a point-by-point manner, with all changes highlighted in blue.

The manuscript by Frei et al. presents a rich database aimed at creating a baseline specific to Denmark for the 87/86 strontium isotope ratio.

Both reviewers welcomed the manuscript but also raised some issues and topics for discussion that need to be addressed by the authors in order to make the manuscript publishable in Plos One.

In particular, reviewer 2 raised some doubts about the basis of the analysis itself. As far as I am concerned, I believe that the manuscript should take into account the reviewers' criticisms, considering them as constructive contributions (even beyond the form). The criticisms should be integrated into the text, also showing possible alternative scenarios with a less assertive and more critical approach. Currently, the manuscript has a dual hybrid dimension, the first being a review paper (without the necessary insights), and the second one being a definition of the baseline (without an effective critical approach).

We would like to clarify that our manuscript is not intended to be read as a review paper. Rather, its primary aim is to present—for the first time—strontium isotope data from non-elite individuals from the Nordic Bronze Age excavated within present-day Denmark. Although this group of individuals is crucial for understanding the socio-dynamics of this formative European period, it has remained largely “invisible” in the archaeological record of the Nordic Bronze Age. Our study provides the first systematic attempt to address this gap in the region of present-day Denmark.

Second, to contextualize these new data, we compiled the first comprehensive database of published strontium isotope measurements from archaeological human remains (tooth enamel and pars petrosa bone) across present-day Denmark (excluding Bornholm). This dataset provides a valuable comparative reference for Sr isotope studies. Based on this compilation, we propose a statistically derived human-based bioavailable strontium isotope range representative for this region. We only propose this approach for this region because:

1. Denmark’s relatively homogeneous, glaciogenic surface geology, which was also exemplified by the resulting relatively narrow surface-water baseline published by Frei and Frei (2011), which suggests that due to this unique geological framework, substantial large-scale variability in bioavailable Sr isotopes are not expected; and

2. the substantial corpus of accumulated, but previously scattered, published human Sr isotope data (summarized in Table S1 with full source references) is now sufficient to support robust statistical analyses of past bioavailable ranges. This combination of relatively homogeneous, glaciogenic surface geology and extensive data coverage creates a unique situation in which a national-level human-based range can be meaningfully defined. We explicitly acknowledge that this approach is only valid under such geological conditions and is not transferable to regions with more heterogeneous geological settings.

We also recognize that our earlier use of the term “baseline” may have caused confusion regarding the nature of this human-based range. We have therefore removed the word “baseline” and now refer solely to a “range.”

Furthermore, unlike traditionally defined baselines or isoscapes, our dataset is not spatially geo-referenced in a way that reflects local environmental conditions at specific geographic locations. In this regard, we fully agree with Reviewer #2. Consequently, our archaeological dataset cannot be used to construct an isoscape for Denmark (e.g., through kriging), and our proposed national range is not intended for assessing intra-Denmark mobility patterns.

In our revised manuscript we have provided interpretations of the new data from the non-elite Bronze Age individuals, in relation to the two previously publblished baselines for Denmark. Furthermore, we compare these interpretations with how the related to the comprehensive dataset in terms of the range. This allows us to distinguish individuals who are “local” (i.e., originating from the region of present-day Denmark excluding the island of Bornholm) from those who are “non-local”. We present the different possible interpretations depending on which previous baselines are used, and how these interpretations compare to the range calculated from the human-based database.

In other words, to avoid further misunderstanding, we have adopted Reviewer #2’s suggestion and refrain from referring to this range as a “baseline.” We instead describe it as a statistically well-defined, human-based bioavailable strontium isotope range, which can be used as an additional reference material alongside existing proxy-based baselines such as the surface-water baseline of Frei and Frei (2011) and the fauna/human-based reference range of Frei and Price (2012). The Supplementary Information also provides a detailed comparison of our human-based range with previously established baselines derived from surface water, soil leachates, and plants.

Finally, we emphasize that the presentation of this human-based range is only one component of the study. The principal objective remains the investigation of mobility (or lack thereof) among non-elite Bronze Age individuals in present-day Denmark—an aspect that has never before been studied in this region.

Beyond these general considerations, I must add a series of notes:

1. The database in Table S1 must also include a column indicating whether the individuals were cremated or inhumed.

We have added two additional columns specifying “non-cremated” (which include inhumations, bog and midden finds, and finds in gallery graves and sacrificial wells) and “cremated”.

2. The database contains values with four or five decimal places: where possible, it would be advisable to standardise the values as much as possible to the fifth decimal place.

S1 Table contains the data as they were published in the respective literature. Some publications only state 4 digits in the 87Sr/86Sr ratios, some do not report errors. Where possible, we now report data to 5 significant digits. This is not possible for cases where data are reported to 4 digits only in the respective source literature cited.

3. I have carried out a quick analysis of the data in Table 6: as can be seen in the attached PDF file, 40.4% of the isotopic ratio values are present in at least two individuals, up to a maximum of six individuals, all with a value of 0.7102, and six with a value of 0.7101. Twenty-three per cent of the duplicates have isotope ratio values to four decimal places, but the remaining 77% are values to five decimal places. Although a certain percentage of duplicate data may be “reasonable” in an analysis of this type, I feel that these percentages are excessive and I invite the authors to further check Table S1. Finally, I think that this situation, if confirmed, should be discussed in the article and justified.

We appreciate the effort in inspecting the data. As mentioned above under point 2, we report data as they were published in the respective literature. Hence, individuals with isotopic values reported only with 4 digits of the 87Sr/86Sr ratios (i.e., ratios that were rounded to the fourth digit) might appear as duplicates, but in reality, they are pseudo-duplicates. However, we are restricted to the reporting of the data as appearing in the respective literature.

4. In my opinion, the use of MAD should be better justified with more literature. Furthermore, the choice to use a multiplier of 2 to define the range of “local” individuals should be better justified (also because the bibliography cited by the authors suggests a value of 2.5). I have taken the liberty of attaching a graph showing the different ranges resulting from different multiplier values. Interestingly, the value of 2.5 seems to be close to the value of Tukey's method, which is the most widely used in the literature for defining outliers.

We appreciate the comment regarding the statistical method used and the effort in illustrating how different MAD parameters, as well as how the Tukey method, affect the results. While we have considered Tukey’s method, we have chosen a MAD approach to delineate outliers in our dataset, as the distribution is very narrow around the median (high kurtosis) with a strong, but flat, positive tail (positive skewness). For this reason, we applied a multiplier factor of 2 in our MAD approach. As described by Leys et al. (2013), this factor is considered “poorly conservative,” yet it effectively captures and delineates outliers in the flat tail as well as slightly asymmetric portions of the main data distribution—particularly elevated ⁸⁷Sr/⁸⁶Sr values—without overestimating outliers. The effectiveness of this approach is reflected in the very low skewness and kurtosis values reported in Table 2, indicating a well-defined Gaussian distribution after outlier removal.

In our study, where we assume that the majority of humans are of local origin (i.e., within present-day Denmark). In this context, the median is an ideal reference. By contrast, Tukey’s method, which uses interquartile ranges, is less robust and could underestimate the number of outliers—particularly for data in the slightly asymmetric portion of the central distribution that would not be classified as outliers using Tukey’s approach.

While a detailed discussion of the advantages and disadvantages of different statistical methods is beyond the main scope of our manuscript, we have added a paragraph in the section entitled “The ⁸⁷Sr/⁸⁶Sr database of archaeological human remains” that elaborates on the reasoning of our choice. In this paragraph, we also added the reference Lightfoot and O’Connell (2016), who recommend using both, MAD and Tukey’s IQR, as a robust approach for identifying migrants in archaeological population samples, based on a global survey of oxygen isotopes in ancient human tooth enamel and bone bioapatite.

PLOS ONE

Journal Requirements:

2. In your manuscript, please provide additional information regarding the specimens used in your study. Ensure that you have reported human remain specimen numbers and complete repository information, including museum name and geographic location.

All but the samples from the two sites from which new data is reported have been published in the literature with the necessary sample information. We found some additional sample specifications for some of the compiled data in the respective literature and added some of the museum numbers lacking in S1 Table where possible.

With respect to the specimens from the new data generated in this study all details are reported herein including the museum numbers, and geographic location.

We have added this disclosure to the acknowledgments.

This study was made possible through the funding to KMF provided by the Carlsberg Foundation “Semper Ardens” advance research grant CF18-0005 for which we are very grateful.

https://www.carlsbergfondet.dk/en

We added this disclosure to the acknowledgments.

We have added this statement to the cover letter.

4. We note that Figure 1 in your submission contain map images which may be copyrighted. All PLOS content is published under the Creative Commons Attribution License (CC BY 4.0), which means that the manuscript, images, and Supporting Information files will be freely available online, and any third party is permitted to access, download, copy, distribute, and use these materials in any way, even commercially, with proper attribution. For these reasons, we cannot publish previously copyrighted maps or satellite images created using proprietary data, such as Google software (Google Maps, Street View, and Earth). For more information, see our copyright guidelines: http://journals.plos.org/plosone/s/licenses-and-copyright.

We have purchased the background map image from iStock and redrafted Figure 1 with this background image. We provide written permission of iStock to publish this map image under the CC BY license in a separate uploaded "Other" file with our submission of the revised manuscript version. We also correctly refer to the source and credit in the respective figure caption.

We added this text in caption of Figure 1.

b. If you are

---

## [Decision Letter · Decision Letter 1]

7 Jan 2026

Bronze Age non-elite mobility in Denmark examined through a new human-based bioavailable strontium isotope range

PONE-D-25-53013R1

Dear Dr. Frei,

We’re pleased to inform you that your manuscript has been judged scientifically suitable for publication and will be formally accepted for publication once it meets all outstanding technical requirements.

Kind regards,

Luca Bondioli, PH.D.

Academic Editor

PLOS One

Additional Editor Comments (optional):

Reviewers' comments:

Reviewer's Responses to Questions

**Comments to the Author**

Reviewer #1: All comments have been addressed

Reviewer #2: All comments have been addressed

2. Is the manuscript technically sound, and do the data support the conclusions?

Reviewer #1: Yes

Reviewer #2: Yes

3. Has the statistical analysis been performed appropriately and rigorously?

Reviewer #1: N/A

Reviewer #2: Yes

4. Have the authors made all data underlying the findings in their manuscript fully available?

Reviewer #1: Yes

Reviewer #2: Yes

5. Is the manuscript presented in an intelligible fashion and written in standard English?

Reviewer #1: Yes

Reviewer #2: Yes

Reviewer #1: (No Response)

Reviewer #2: The authors have adequately addressed the comments. The transition from calling the data a "baseline" to a "range" is a critical improvement. It correctly frames the findings as a statistical summary of typical Danish signatures rather than a spatially explicit environmental model. The revised manuscript is technically sound and provides a valuable new resource for the archaeological community.

**Do you want your identity to be public for this peer review?** For information about this choice, including consent withdrawal, please see our Privacy Policy

Reviewer #1: No

Reviewer #2: No

---

## [Editor Report · Acceptance letter]

PONE-D-25-53013R1

PLOS One

Dear Dr. Frei,

I'm pleased to inform you that your manuscript has been deemed suitable for publication in PLOS One. Congratulations! Your manuscript is now being handed over to our production team.

Kind regards,

on behalf of

Dr. Luca Bondioli

Academic Editor

PLOS One